# Histopathological Variants of Cutaneous Neurofibroma: A Compendious Review

Neha S. Nagrani and Jag Bhawan *

Dermatopathology Section, Department of Dermatology, Boston University School of Medicine, Boston, MA 02118-2415, USA
* Correspondence: jbhawan@bu.edu

**Abstract:** The first description of histopathological variants of neurofibroma dates back to 1994. Over the years, many individual case reports elucidating unusual histologic features in neurofibroma have been added to the literature, some of which have defined criteria, with the others falling under the roof of benign neural neoplasms. These unusual features, which sometimes may lead to pauses in identifying a common benign tumor such as neurofibroma. Awareness of these variants may help dermatopathologists avoid misinterpretation. Thus, this review aims to summarize all novel and unusual histopathological variants of cutaneous neurofibroma reported to date, in addition to any unusual variants that we encountered in our practice.

**Keywords:** cutaneous neurofibroma; histopathological variants; classification





## 1. Introduction

Neurofibroma has been a subject of intense medical curiosity from ancient Egypt until the present date. Before the word 'neurofibroma' was introduced, in the late 1700s and early 1880s, the term 'cutaneous fibromas' and 'neuromas' were used to describe skin tumors and deep nerve tumors in patients with neurofibromatosis, respectively. Although several researchers have made remarkable contributions to the understanding of this disease, Von Recklinghausen is credited with discovering the pathogenesis of these tumors. The nature of the neoplastic tissue in both types of tumor was described to be almost identical and result from the mingling of neural elements and connective tissue cells, thus giving birth to the term 'neurofibroma' [1–3].

Neurofibroma (NF) is defined as a histologically benign (WHO grade I) peripheral nerve sheath tumor composed of cells of diverse lineage including Schwann cells, fibroblasts, perineurial cells, mast cells and macrophages with intermixed axons [4]. It is conjectured that Schwann cells constitute the neoplastic component, and the other cells are either present at the start or recruited eventually into the lesion. However, to date, the definitive cell of origin of NF remains ambiguous. Historically, Von Recklinghausen believed these tumors were of mesenchymal origin. It was not until 2002 that Schwann cells were undoubtedly shown to be the origin of NF. Current models indicate that different types of NF arise from a common cell of origin arising from a subpopulation of migrating neural crest stem cells (NCSCs) [5].

NF can occur sporadically or in association with genetic syndromes such as neurofibromatosis type 1 (NF1). Several efforts have been made to classify NF based on their clinical appearance and histologic features, leading to multiple classification systems being published in the literature and some unpublished systems presented at the European NF meeting (2008) by Ortonne et al. [6]. The basic theme proposed in these systems involved classifying NF based on its anatomic location, growth pattern, and relationship to nerve and pathogenesis (Table 1).

**Table 1.** Classification of neurofibroma.

| | |
|---|---|
| Anatomic location | Cutaneous<br>Cutaneous/subcutaneous<br>Deep |
| Growth Pattern | Localized<br>Diffuse/infiltrating |
| Relationship to nerve | Intraneural<br>Extraneural |
| Pathogenesis | Endoneurial<br>Perineurial<br>Epineurial |

Cutaneous NF (cNF) can be further categorized as localized or diffuse. The localized variant is the most common, usually presenting in adulthood as an asymptomatic solitary polypoid or nodular lesion arising anywhere in the body without particular anatomic distribution and is rarely associated with NF1. However, the presence of multiple such lesions raises concerns regarding NF1. Diffuse cutaneous neurofibroma is an uncommon and clinically distinctive variant that usually presents in young adults as an ill-defined plaque of dermal and subcutaneous thickening with overlying hyperpigmentation, commonly on the head and neck or trunk. Approximately 10% of cases are associated with NF1. Localized and diffuse cNF rarely progress to malignant peripheral nerve sheath tumors (MPNST) [4].

In patients with NF1, emphasis has been placed on distinguishing cNF, which is limited to dermis from the extension of deep NF into the skin, especially in cases demonstrating atypical features such as cytological atypia or increased cellularity. It is believed that the atypical changes in cNF represent reactive or degenerative changes while similar changes in a deep NF can be a manifestation of malignant transformation [6].

NFs, irrespective of their location, share common histologic and immunohistochemical features. However, unusual histologic features have been described in the literature occasionally, some of which have posed difficulty in its diagnosis as well as in differentiating it from other tumors. Megahed outlined ten different histopathological variants of NF [7]. Since then, over the years, many individual case reports elucidating peculiar histologic findings in cNF have been added to the literature. Awareness of these variants may help dermatopathologists avoid misinterpretation and make the correct diagnosis of cNF. Thus, this review aims to summarize all novel and unusual histopathological variants of cNF reported to date, in addition to unusual variants that we encountered in our practice.

## 2. Conventional/Classic cNF

Classic localized cNF is a circumscribed unencapsulated tumor usually located in the dermis, often with extension into the subcutis. It is primarily comprises only endoneurial components, specifically lacking both an intact perineurium and intact epineurium [8]. It is composed of proliferations of multiple cell types in a variably collagenous and myxoid stroma. Besides Schwann cells, which are the predominant cell type, it contains perineurial-like cells, fibroblasts, endoneurial fibroblasts, mast cells, macrophages, endothelial cells and pericytes with scattered, intermingled axons (Figure 1A).

cNF shows a variety of appearances depending on the relative proportions of its constituent cellular components and variable stroma. The various histopathological variants of a cNF are enumerated in Table 2. The most typical form consists of haphazard and loosely arranged spindle cells with poorly defined, pale eosinophilic cytoplasmic processes and elongated wavy or buckled hyperchromatic nuclei mixed with a population of short spindle cells, mast cells and occasional nerve trunks in a pale staining fibrillar, collagenous and sometimes myxoid stroma (Figure 1B) [9]. Often the Schwann cells are arranged in short bundles with dense collagen, referred to as the "shredded carrot pattern" [9]. Occa-

sionally, the cells may be arranged in short fascicles, whorls, or even in a storiform pattern. Small blood vessels are admixed with the neural components. The lesion may extend around the adnexa without destroying them (Figure 1C). A single case report demonstrates the presence of abundant eosinophils. Eosinophilic chemotaxis influenced by mast cells has been suggested [10]. The presence of abundant lymphocytic aggregates has been documented [11].

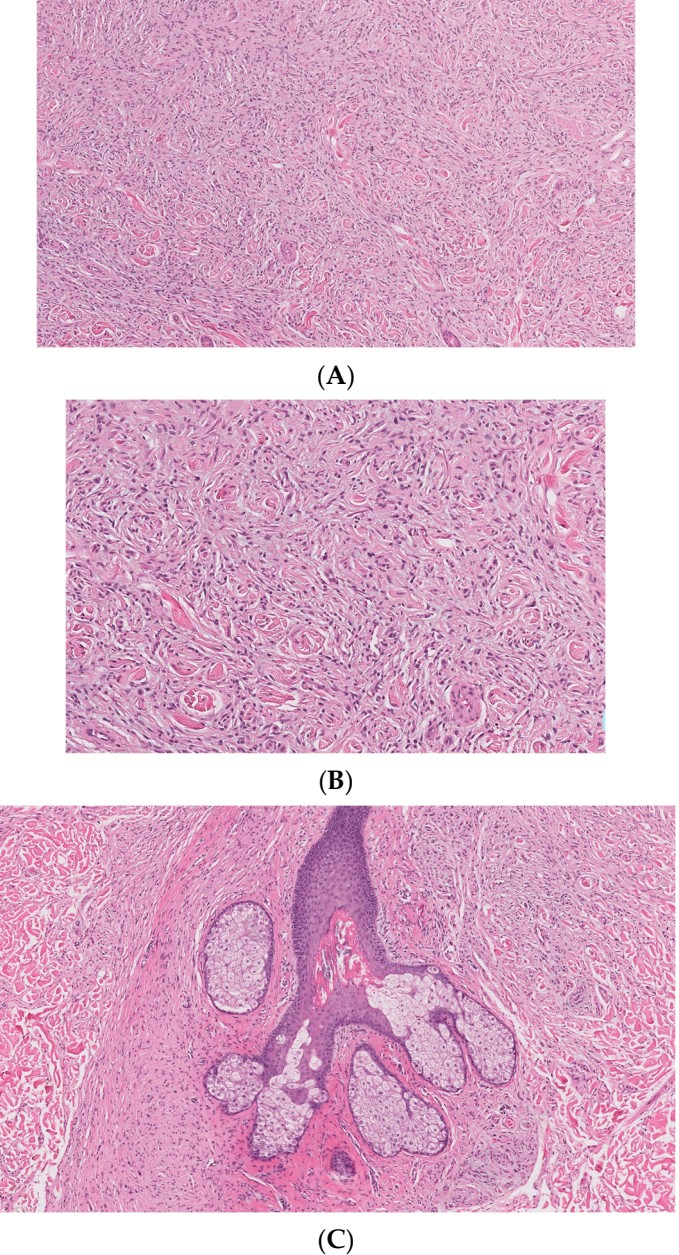

**(A)**

**(B)**

**(C)**

**Figure 1.** Conventional cNF. (**A**) The lesion is made of scattered spindle cells that correspond to Schwann cells, fibroblasts and mast cells admixed numerous capillaries (Hematoxylin & eosin (H&E) stain, 50×) (**B**) (H&E, 100×). (**C**) Lesional cells surrounding the pilosebaceous unit (H&E, 40×).

Table 2. Histopathological variants of Cutaneous Neurofibroma.

| Conventional/Classic cNF |
| --- |
| According to variations in cell morphology: <br> • Epithelioid; <br> • Granular cell; <br> • Balloon cell/clear cell; <br> • Dendritic cell NF with pseudorosettes; <br> • Floret-like multinucleated giant cells; <br> • Meissnerian; <br> • Pacinian; <br> • Pigmented/melanotic; <br> • Atypical; <br> • Cellular; <br> • Microcystic pseudoglandular. |
| According to the variations in stroma: <br> • Myxoid; <br> • Hyalinized; <br> • Sclerotic; <br> • Lipomatous; <br> • Angioneurofibroma. |

The heterogeneous cellular composition of NF is highlighted on immunohistochemistry; only a portion of cells are S-100 positive, giving a somewhat "spotty or loose" staining pattern admixed with CD34-positive fibroblasts and epithelial membrane antigen (EMA)-expressing perineurial cells with scattered axons highlighted by neurofilament protein [9]. CD34 immunoreactivity in NF show a distinct "fingerprint" pattern. The "fingerprint" is due to the positive staining of endoneurial fibroblasts or perineurial-like cells in between the collagen bundles in a whorled configuration, resembling a human fingerprint [12,13]. This pattern, if present in more than 60% of lesions, favors cNF and helps to distinguish it from early desmoplastic melanoma [14] However, it is not specific or sensitive, as it can be seen in perineurioma and dermatofibrosarcoma protruberans, and hence needs to be interpreted in the appropriate clinical context.

Diffuse cNF has a similar cytomorphology to localized cNF but lacks its well circumscribed nature and shows a diffuse infiltrative pattern of cells that permeate into the surrounding adipose tissue and skeletal muscles.

### 3. Histopathological Variants

*3.1. According to the Variations in Cell Morphology*

3.1.1. Epithelioid cNF

Epithelioid cNF is a rare variant of cNF, which is mentioned in books by Weiss (3rd edition) and Mckee [15,16]. In the literature, a single lesion showing epithelioid morphology has been described by Megahed [7]. Clinically, it presents as a solitary lesion with a buttonhole sign, indistinguishable from classical cNF. It may or may not be associated with NF1.

Epithelioid cNF, as the name suggests, demonstrates predominant areas of epithelial-like cells arranged in nests or short cords in the background of classical cNF. These cells have abundant eosinophilic cytoplasm, indistinct cell borders, and oval, round, fusiform or vesicular nuclei, often with degenerative changes including intranuclear pseudoinclusions. Binucleate and multinucleated cells may be present. Occasionally, epithelioid cNF may show large, hyperchromatic and atypical nuclei [7]. The stroma may be myxoid or vaguely chondroid, but in many cases the stroma is heavily collagenous [9]. The epithelioid cells are S-100 positive.

An umbrella term "benign epithelioid peripheral nerve sheath tumor" (BPENST) was proposed in a landmark study by Laskin et al. to encompass all benign peripheral nerve sheath tumors with predominance of epithelioid cells [17]. This included epithelioid cNF,

epithelioid Schwannoma and BEPNST of indeterminate histogenesis, which are difficult to subtype into either of the first two categories. The study indicated that epithelioid cNF acts in a benign fashion, which was evident from the absence of recurrence and metastasis. Overall, the significance of this epithelioid morphology is still not known [17].

### 3.1.2. Granular Cell Cant

The term granular cell variant of cNF was historically mentioned by Megahed and in a case report by Finkel et al. in 1982 [7,18]. It has been described as having a similar architecture to classical cNF but cells in focal areas show numerous PAS-positive, diastase-resistant granules of various sizes in the cytoplasm. However, with the dawn of immunohistochemistry, these tumors are now regarded as granular cell tumors [18].

### 3.1.3. Balloon Cell/Clear Cell cNF

Clear-cell changes in a cNF have reported in only two cases to date and none of them were associated with NF1. Balloon-cell or clear-cell change refers to the presence of collection of abundant, markedly swollen, vacuolated cells with clear cytoplasm in the background of classic cNF (Figure 2A,B). The nuclei are round, located both centrally and eccentrically, not scalloped and contain prominent nucleoli. These clear cells are intimately associated with spindle cells, with wavy nuclei showing areas of transition. The clear cells are S-100 positive and negative for MART-1 and CD68, differentiating them from balloon cell nevus and granular cell tumor, which are the closest differentials [19]. The other differential diagnoses include clear cell fibrous papule, balloon cell melanoma, chordoma, metastatic conventional renal cell carcinoma, perivascular epithelioid cell tumor and clear cell dermatofibroma. Clear cell fibrous papules contain monomorphous clear cells with fibroblasts and ectatic capillaries Clear cells are negative for S-100 protein and positive for NKI/C3 and CD68. Balloon cell melanoma also displays cellular pleomorphism and stains for Mart-1, S-100, Sox-10 and HMB-45. [20]. Chordoma contain characteristic clear physaliferous cells, which are large polygonal cells with central nuclei and vacuolated cytoplasm and are within a mucinous matrix. The cells stain with S-100, vimentin, cytokeratins, and EMA. Perivascular epithelioid cell tumors express HMB-45, Melan-A, MiTF, and desmin or smooth muscle actin and are negative for S-100, EMA, and pancytokeratin.1. The clear cells in metastatic renal cell carcinoma are positive with cytokeratin, vimentin, CD10, and RCC-Ma. Clear cell dermatofibromas contain spindled fibrohistiocytic cells and clear cells which express CD68 and CD10 and are negative for S100. Clear-cell change is postulated to be a degenerative change [19].

### 3.1.4. Dendritic Cell cNF with Pseudorosettes

Dendritic cell cNF with pseudorosettes (DCNP) gets its name from the characteristic morphology and architectural arrangement of the cells. In 2001, Michal et al. reported 18 distinct cutaneous nerve sheath tumors and proposed the term DCNP [21]. Since then, close to 40 cases have been reported [22].

Clinically, DCNP presents as a solitary, flesh colored, protuberant or dome shaped nodule in adults with wide anatomic distribution. Histopathologically, it is a well-circumscribed tumor located in the dermis, with the long axis of the tumor running perpendicularly to the epidermis; this tumor can infiltrate deeply into the subcutis, although this is rate. The deeper part of the lesion often demonstrates a multinodular arrangement. It is composed of two main cell types (Figure 3). Type I cells are small, dark cells that vaguely resemble lymphocytes with slightly irregular or cleaved nuclei, containing small amount of inconspicuous cytoplasm. Type II cells are larger and pale staining with vesicular nuclei and abundant eosinophilic cytoplasm forming dendritic extensions. Type I cells concentrically surround type II cells to form distinctive pseudorosettes. Both cell types stain positively for S-100 and CD57, although this is more conspicuous in type II cells, suggesting a neural differentiation with a dermal interstitial dendritic cell lineage [21].

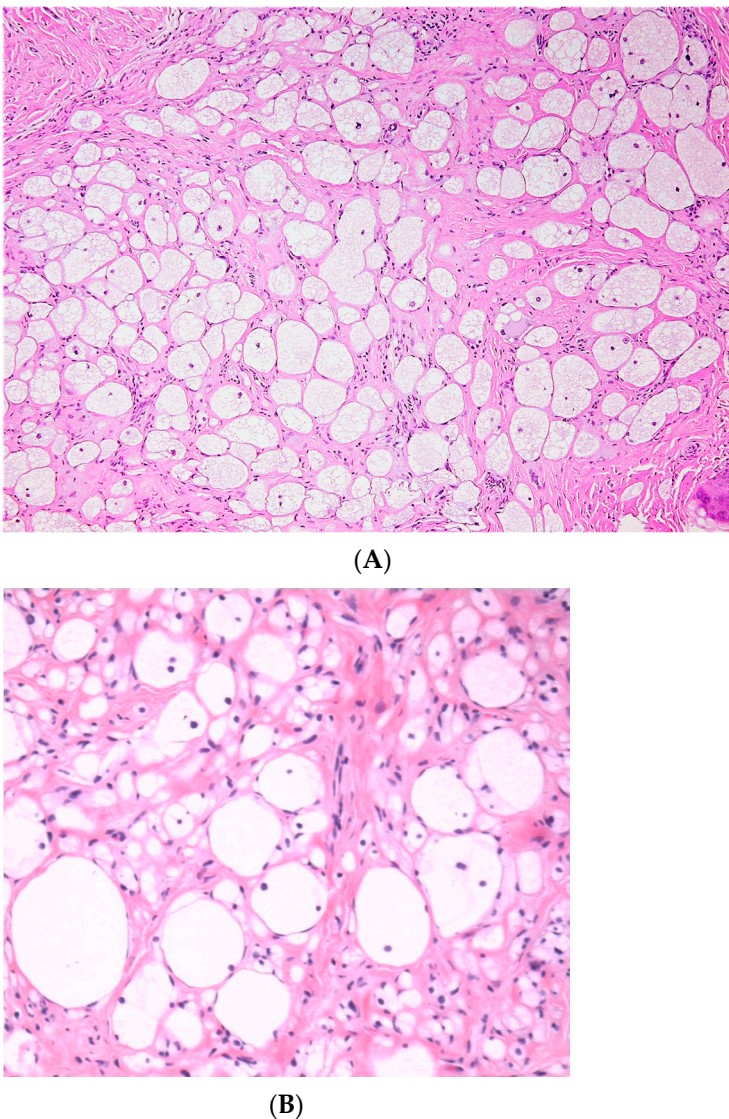

**Figure 2.** Clear/Balloon cell cNF (**A**) Collection of large clear vacuolated cells, H&E, 200× (courtesy: Dr. Tamie Ferringer) (**B**) Cells with from clear to granular cytoplasm, round nuclei and prominent nucleoli, H&E, 400× (courtesy: Dr. Wilsher).

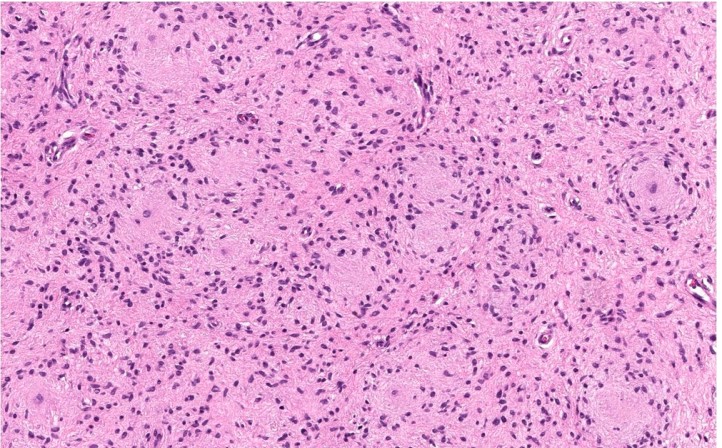

**Figure 3.** Dendritic cell cNF with pseudorosettes: pseudorosettes with type I cells concentrically surrounding type II cells H&E, 100×.

Woodruff et al. suggested that this tumor may be an atypical type of melanocytic nevus with neural differentiation [23]. This was dismissed with the immunohistochemical demonstration of CD57 supporting neural origin [24]. Saggini et al., in their case report, noted striking architectural similarities between DCNP and solitary circumscribed neuroma (SCN), which therefore suggested that DCNP is a histologic variant of SCN [25]. This needs further molecular support.

DCNP cases have not been associated with NF1 to date. There is only one reported case with two lesions and one large cafe-au-lait spot that did not fulfil the diagnostic criteria for NF1 [26]. DCNP has a benign clinical course with no reports of recurrence or malignancy [22].

### 3.1.5. Floret-Like Multinucleated Giant Cells

Floret-like multinucleated giant cells (FMGCs) have been commonly reported in soft-tissue tumors such as pleomorphic lipoma, giant cell fibroblastoma, giant cell collagenoma and giant cell angiofibroma/solitary fibrous tumor [27]. Magro and colleagues first reported the presence of numerous FMGCs in a diffuse type cNF from a patient associated with NF1 [28]. Subsequently, the same author conducted the largest clinicopathologic study of 94 cases of sporadic and NF1 associated cNF of which 15 cNF demonstrated FMGCs [29]. Among cNF, only the diffuse type contained FMGCs. FMGCs in sporadic and NF1-associated cNF were morphologically identical. There was no statistically significant association found between NF1 and FMGCs [29], contrary to the suggestion by Taungjaruwinai and Goldberg [30].

FMGCs are medium-to-large cells exhibiting multiple nuclei, arranged either randomly or in a wreath like configuration. The nuclei have vesicular chromatin with prominent nucleoli. The cells have abundant eosinophilic cytoplasm with dendritic extensions. FMGCs are scattered throughout the lesion of classical cNF (Figure 4). They are immunoreactive for vimentin, CD34 and are negative for S-100, CD68, EMA and cytokeratins. FMGCs differ from the multinucleated cell identified in cellular cNF and atypical cNF, which may have cytologic atypia, lack a floret-like appearance and are S-100 positive and CD34 negative. A single case report showed positive expression of CD68 in FMGCs [31]

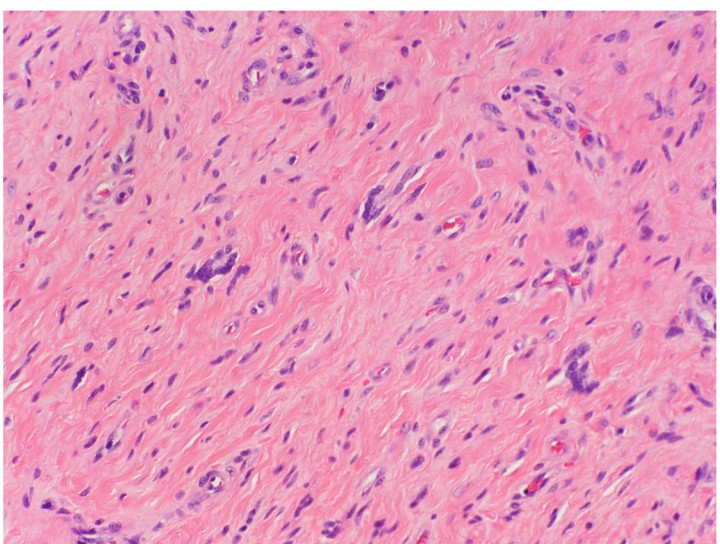

**Figure 4.** cNF showing wreath like multinucleated giant cells, H&E, (400×) (courtesy: Dr. Lynne Goldberg).

The pathogenesis of FMGCs in a cNF is unclear. Some authors consider them to be a morphological reactive change in the native dermal or endoneurial fibroblasts or dendritic cells in response to trauma, hypoxia or an unknown stimulus or even a reparative change, while others suggest the potential role of mast cells. A study observed a significant association between the presence of FMGCs and large size of the cNF. The possible association

between FMGCs and the long-term risk of neoplastic changes in NF1 patients has yet to be studied [32].

### 3.1.6. Meissnerian cNF

Meissnerian cNF is a term proposed by Sode et al. in 2020, to describe a novel variant of cNF demonstrating a dominant pattern composed of structures resembling Wagner-Meissner corpuscles (also known as pseudo-meissnerian bodies or Meissner corpuscle-like or tactile corpuscle-like bodies) with features of classical cNF at the periphery [33]. The presence of focal or sparsely scattered tactile corpuscle-like bodies in NF is well documented in diffuse and plexiform types [34,35].

Clinically, this lesion presents as an asymptomatic soft skin-colored papule in a patient with no history of NF1. Histopathologically it is a well-circumscribed, unencapsulated dermal tumor composed of tightly packed structures containing pink, fibrillary material, and peripheral cells with small, wavy nuclei, resembling Meissner corpuscles, constituting more than 90% of the tumor. The periphery of the tumor is made up of a mixture of short, spindled cells and interspersed mast cells. Pseudo-meissnerian bodies were positive for S-100, Sox-10, collagen IV and CD56. Neurofilament highlighted axons within the center of the pseudo-meissnerian bodies.

The pathomechanism underlying formation of Meissner corpuscle-like bodies is still unclear. Some investigators (Masson 1970; Schochet and Barrett 1974; Lassmann et al. 1977), based on the light microscopic features, have considered them to be of Schwann cell origin, whereas others (Weiser 1975; Smith and Bhawan 1980) have emphasized their morphological similarities to perineurial cells, as demonstrated under electron microscopy [35]. Based on immunohistochemical and electron microscopic findings, Watabe et al. indicated them to be Schwann or Schwann-related cells [36]. The clinical course of Meissnerian cNF is believed to be benign, similar to classical cNF.

### 3.1.7. Pacinian cNF

Pacinian cNF is characterized by the predomination of components resembling Pacinian corpuscles within the classical cNF tumor [37]. Initial descriptions of Pacinian cNF date back to 1894 by Thoma, and later were given by Prichard and Custer in 1952, as well as by Prose et al., in 1957 [38]. Over the years, there has been considerable variation in its description. It has been described by various other terms, such as plexiform neurofibroma with aberrant tactile corpuscles, benign fibromyxoma, Pacinian neurofibromatosis, neurofibroma plexiform, myxoid neurofibroma, bizarre cutaneous neurofibroma, and nerve sheath myxoma; some of these are used synonymously while the others are misnomers or represent other tumors [39].

Pacinian cNF usually presents as an occasionally painful, palpable mass, commonly seen on fingers and palms. It can be sporadic or associated with NF1. Histopathologically, it is a well-demarcated dermal tumor, often extending to subcutaneous fat composed of variably sized round or ovoid lobules, each showing a central homogeneous, hypocellular, eosinophilic core surrounded by as many as 30 pale-staining, concentric collagenous lamellae resembling Pacinian corpuscles (Figure 5) [39]. The superficial corpuscles tend to be smaller (4–7 lamellae), and deeper immature corpuscles are larger. The lobules contain elliptical or spindle-shaped nuclei in both central core and the surrounding lamellae, and the concentric lamellae merges with the collagen fibers of adjacent dermis. The adjacent tissue may be cellular and contain poorly formed nerve bundles. The stroma may be mucinous. The clinical course is benign; however, recurrence is reported in 50% of cases.

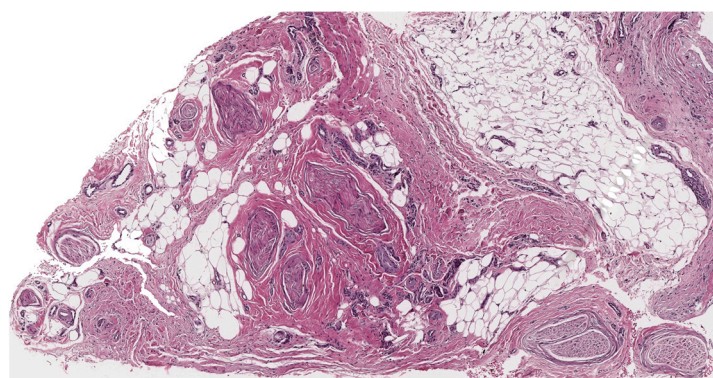

**Figure 5.** Pacinian cNF, clusters of spherical bodies resembling rudimentary Pacinian corpuscles are embedded in mature adipose tissue, H&E, 10× (courtesy: Dr. Raj Singh, www.PathPresenter.net accessed on 1 October 2022).

### 3.1.8. Pigmented cNF

Pigmented (melanotic) cNF is a rare variant characterized by the presence of pigmented cells within the conventional cNF. The pigmented cells may have a spindled, dendritic, or epithelioid appearance and contain coarse, granular, dark brown pigments in the cytoplasm (Figure 6A). The pigment is highlighted by Fontana–Masson stain indicating it to be melanin. On immunohistochemical stains, they express S-100 and melanocytic markers such as Melan-A and HMB 45, confirming a melanocytic phenotype (Figure 6B) [9]. Non-pigmented cells show mixture of cells positive for S-100, CD34 and vimentin. Focal areas may show a storiform pattern, leading to confusion with pigmented Dermatofibrosarcoma protruberans (DFSP); however, non-pigmented cells in DFSP show no immunoreactivity for S-100. Electron microscopic studies show numerous electron-dense melanin granules in the cytoplasm of pigmented cells [40].

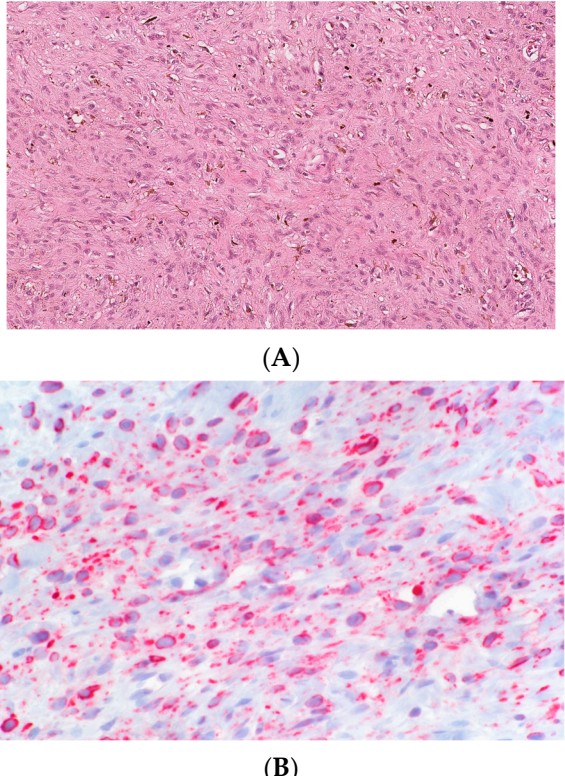

(**A**)

(**B**)

**Figure 6.** Pigmented cNF (**A**) Haphazardly arranged oval to spindle cells containing dark crown pigment, H&E, 100×; (**B**) positive for Mart-1, (immunohistochemical stain, 200×).

Pigmented cNF may or may not be associated with NF1. The pigmented areas have been reported in NFs of different histological differentiation, as in diffuse, combined diffuse and plexiform, combined diffuse and intraneural epithelioid. However, it is not described in localized cNF [40–42]. It is unclear whether the pigmented cells are true melanocytes or Schwann cells with aberrant melanocytic differentiation, as both arise from the neural crest [9]. Some believe that these tumors contain cells fully committed to both lines of differentiation, as well as some "transitional" cells, bridging the morphologic and immunohistochemical spectrum [41].

Considering it to be a melanocytic differentiation, some authors have given the name 'Melanocytic NF' to describe some non-pigmented tumors containing cells that express melanocytic markers surrounded by NF-like cells. It is believed that the diffuse variant of NF is more likely to present such differentiations [43].

Pigmented cNF is believed to have a benign, non-aggressive clinical course but a thorough examination of the specimen is essential to exclude the rare simultaneous occurrence of both malignant melanoma and malignant transformation, especially in plexiform tumors in patients with NF1 [40,41].

### 3.1.9. Atypical cNF

Nuclear atypia is known to be present in some sporadic and NF1-associated NFs. Usually, it is reported in diffuse and plexiform NF but has been described in cNF as well. Atypical cNF is histologically defined as the presence of focal or more pronounced areas of cells with nuclear atypia and occasional mitoses in the background of classical cNF [44] Such cells may constitute 5–50% of the total cellularity [45]. On low-power examinations, the lesion may exhibit a lamellar, fibrillar or focal fascicular growth pattern (Figure 7A). On high-power examinations, the cells show nuclear atypia in the form of nuclear enlargements of 2–3-fold or more, hyperchromasia with irregular chromatin distribution, intranuclear pseudoinclusions, multinucleated or bizarre forms (Figure 7B). Mitosis is usually absent.

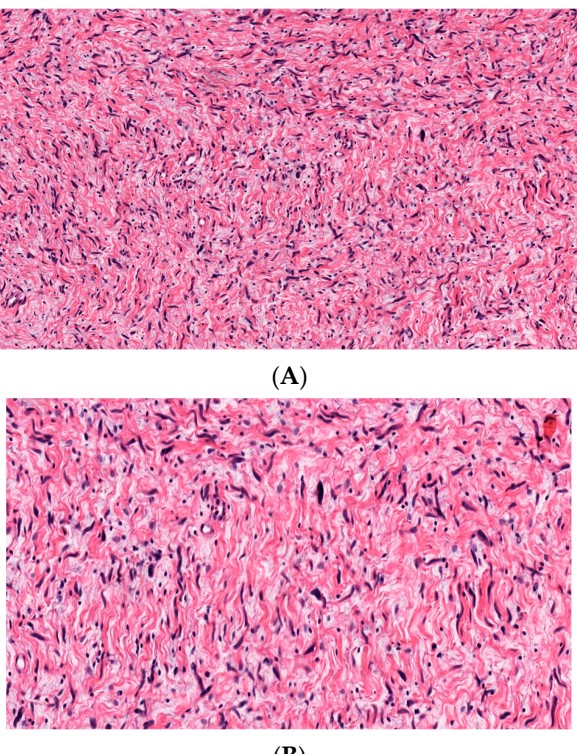

(**A**)

(**B**)

**Figure 7.** Atypical cNF (**A**) Banal and atypical cells are seemingly haphazardly arranged in a collagenous stroma, H&E, 100× (**B**) The atypical cells have hyperchromatic irregular nuclei. H&E, 200× (courtesy: Dr Raj Singh, www.PathPresenter.net accessed on 1 October 2022).

The presence of scattered bizarre nuclei in the absence of hypercellularity, mitotic activity, or loss of neurofibroma architecture has been designated as "degenerative atypia", similar to that in "ancient schwannoma" and other benign neoplasm and has no clinical significance [46]. Atypical cNF refers to an NF showing some worrisome features but not enough to be malignant. There are no scientific criteria to date that can be used to clearly distinguish "degenerative atypia" from the "real atypia" of malignancy or premalignancy. Sporadic cutaneous atypical NFs described in the literature have not shown to recur or undergo malignant change. However, in the NF1 setting, the clinical behavior is still debatable and requires complete excision and careful follow up [44].

The consensus report proposed the term 'Atypical Neurofibromatous neoplasms of uncertain biologic potential' (ANNUBP) for neurofibromatous tumors showing at least two of these features: nuclear atypia, hypercellularity, variable loss of classical neurofibroma architecture and/or mitotic activity beyond isolated mitotic figures (>1/50 high power fields (HPF) and less than 3/10 HPF). This does not represent a distinct diagnostic entity but rather a clinical situation, which requires additional sampling, clinical correlation, and possibly an expert pathology consultation or clinical follow-up for resolution [46].

On immunohistochemistry, diffuse S-100 and SOX10 positivity, CD34 positive fibroblastic network, low Ki-67 proliferation index (<2–5%) and the absence of p53 staining favor ordinary or atypical cNF rather than malignant peripheral nerve sheath tumors (MPNST) [46].

### 3.1.10. Cellular cNF

Cellular neurofibroma is a term used to indicate NFs exhibiting hypercellularity without mitotic activity, cytologic atypia, or loss of neurofibroma architecture. There are no definitive data on the risk of its progression to MPNST. However, if accompanied by other worrisome features, ANNUBP should be considered [46].

### 3.1.11. Microcystic (Pseudoglandular) cNF

Microcystic pseudoglandular cNF is an extremely rare variant of NF. Only two cases have been reported to date. The first case was described by Gomez-Mateo et al. in a 33-year woman with no family history of NF1 [47]. Clinically, it presented as a solitary asymptomatic nodule over the left scapular region. In 2021, Pak and colleagues reported another case in a 23-year-old woman, with no association with NF1, which presented as a slow-growing but painful solitary dome-shaped papule over cheek [48].

Histolopathologically, a well-demarcated unencapsulated dermal tumor is observed, composed of areas of microcystic gland-like spaces lined by a single layer of flat cells interspersed with spindle shaped cells embedded in a matrix of myxoid to collagenous areas. Scattered lymphocytes, mast cells and blood vessels are noted. Stromal spindle cells and cells lining the microcystic spaces stain positive for S100 and SOX-10. These cells are negative for EMA, cytokeratin and CD31. EMA highlights the perineurial cell, outlining the tumor. Both lesions were benign, without the features of nuclear pleomorphism, mitotic figures or necrosis.

Microcystic pseudoglandular pattern should be differentiated from the true glandular component in schwannomas. This variant of NF has only a glandular pattern, where even the gland forming cells are S100-positive, whereas the glands in glandular schwannoma the gland lining cells are positive for cytokeratin lying within a matrix of spindle cell population positive for S100), which demonstrate cytokeratin immunoreactivity and entrapped adnexal structures and, in contrast, have a myoepithelial cell lining. The formation of microcysts is considered to be a degenerative change (Figure 8) based on the low proliferative index within these areas and its mucinous component, which is similar to Schwannoma.

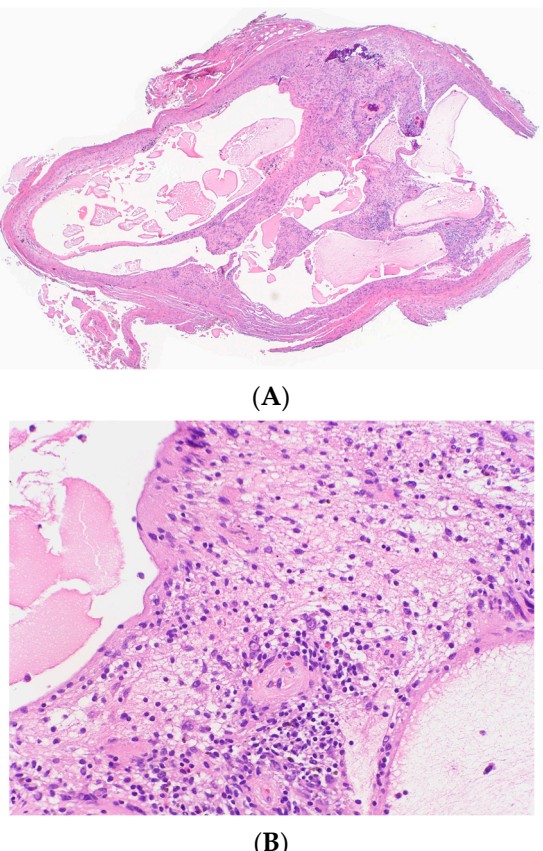

(**A**)

(**B**)

**Figure 8.** Large cystic spaces containing mucinous materials without any epithelial lining in a cystic NF (**A**) H&E, 20×, (**B**) H&E, 200×.

### 3.2. *According to Variation in Stroma*

### 3.2.1. Myxoid cNF

Myxoid cNF represents a classical NF with the extensive deposition of stromal mucin (Figure 9). Sometimes, vacuolated histiocytes, sometimes termed muciphages or "pseudolipoblasts", may be present. It usually presents as a solitary, asymptomatic, flesh-colored to pink or blue, dermal nodule. Higher incidence has been reported in teenagers and young adults and the most common locations are the face, shoulders, and arms [49]. Megahed et al. reported two cases of myxoid cNF in middle aged adults located on the back and retroauricular [7]. It has also been reported in periungual and subungual locations [49,50].

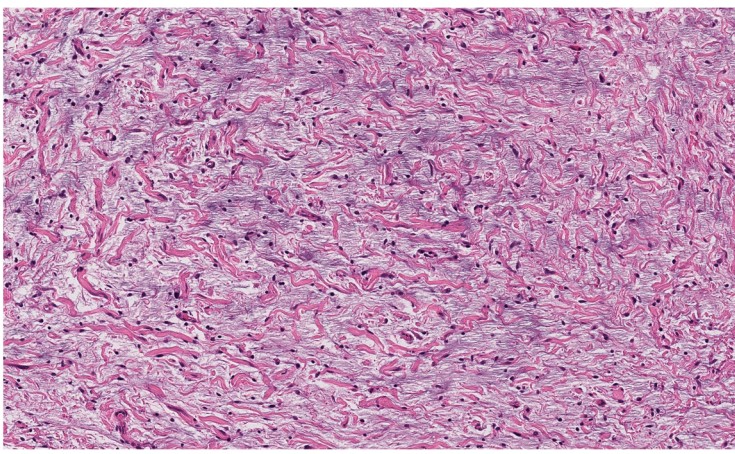

**Figure 9.** Myxoid cNF, neoplastic spindle cells in abundant mucinous matrix H&E 100×.

In the past, myxoid cNF has been described by many different names, including nerve sheath myxoma, neurothekeoma, bizarre cutaneous NF and lobular neuromyxoma. It may or may not be associated with NF1 [9]. It is a component of the NAME syndrome (nevi, atrial myxoma, myxoid neurofibroma, and ephelides). The pathogenesis of myxoid change remains unclear. It has a benign clinical course, and malignant transformation has not been reported [51].

### 3.2.2. Hyalinized cNF

Megahed et al. described hyalinized cNF as a variant of classical NF showing thickened collagen bundles [7]. McHugh et al., in their case series of hyalinized cNF in the skin of female breast, used the criteria of thick collagen bundles in fascicles and/or whorls constituting at least 75% of the tumor [52]. In this case series, it is reported to have an increased association with NF1. the interaction between mast cells and fibroblasts was implicated in the pathogenesis of hyalinized cNF.

### 3.2.3. Sclerotic cNF

The sclerotic cNF is an extremely rare variant of classical NF showing marked sclerosis (Figure 10). There are thick collagen bundles, chiefly arranged in an interweaving pattern with or without prominent clefts, and the tumor cells are scant [53,54] Sclerotic cNF can be purely sclerotic or mixed with the adjacent area of classical neurofibroma. Clinically, it presents as a solitary, hard, nontender nodule. One case of segmental neurofibromatosis has been reported [55]. The positive immunohistochemical staining of tumor cells with S100 helps to differentiate it from sclerotic fibroma and sclerosing perineurioma. Nakashima et al. have hypothesized that mast cells play a role in the pathogenesis of sclerosis [53].

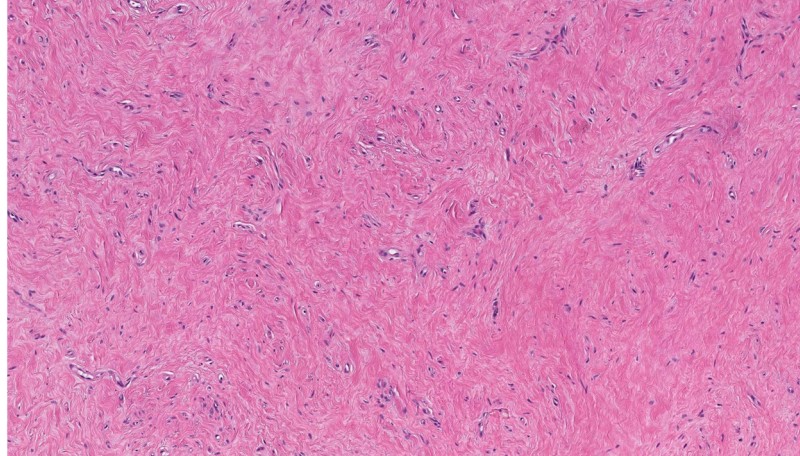

**Figure 10.** Sclerotic cNF, spindle-shaped cells with wavy cytoplasm and elongated nuclei embedded in markedly dense sclerotic stroma, H&E, 20×.

### 3.2.4. Lipomatous cNF

Lipomatous cNF refers to the presence of regularly distributed collections of mature fat cells within a classic neurofibroma (Figure 11). This peculiar variant was first documented by Val-Bernal et al. in 2002 [56]. In the following years, he further described 22 dermal NFs with intratumoral fat [57]. Lipomatous cNF was defined as a tumor that had more than four mature adipose cells tintimately associated that were with the spindle cells. Intratumoral fat was divided into two groups based on its presence: (a) focal and (b) diffuse (regularly interspersed). Eighteen out of the 22 reported NF (5.6%) presented single or grouped mature adipocytes that were focally intermingled with the spindle cells, and four of these tumors belonged to three patients with NF-1. The other four (1.3%) NFs showed a diffuse pattern, with spindle cell proliferation with regularly scattered adipocytes comprising at least 30% of the tumor. Adipocytes in all these tumors had no connection to the underlying

subcutaneous fat. Floret-like multinucleated giant cells have also been documented in lipomatous cNF [32,58].

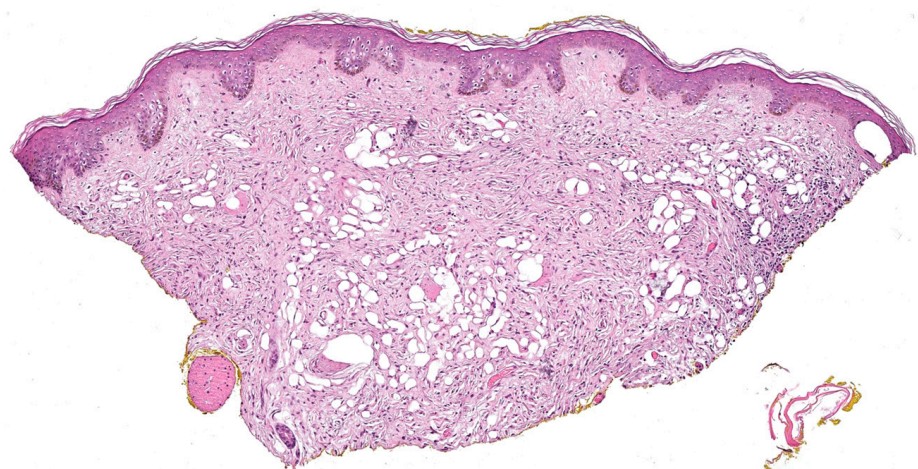

**Figure 11.** Lipomatous cNF, lesion showing interlacing fascicles or whorls of spindle cells focally intermingled with adipose tissue., H&E, 20×.

In a systematic review by Rozza-de-Menezes et al. in 2018, no statistically significant difference was found in the prevalence of sporadic and NF1-associated lipomatous cNF. The pathogenesis of lipomatous differentiation in cNF remains obscure. Various studies have attributed focal lipomatous differentiation to a metaplastic or degenerative change. Others have suggested that the adipocytes originate from the differentiation of local multipotent neural crest stem cells after emigration in the skin [57–60]. There is no evidence of any recurrence or neoplastic change in lipomatous cNF at present.

### 3.2.5. Angioneurofibroma

Malignant tumors are usually highly vascular and are dependent on vascular supply for growth. In contrast, the angiogenic potential of benign tumors other than vascular tumors is not adequately studied. Arbiser et al., in his study of NF from patients with NF1 and sporadic NF, demonstrated high vascular density in NF as compared to normal skin [61]. There was increased vascular endothelial growth factor immunoreactivity, suggesting their angiogenic potential. However, studies by Friedrich et al. have shown variable vascular density in cutaneous and plexiform neurofibroma associated with NF1 that is not significantly different from normal human skin [62].

Saxer-Sekulic et al., in his case series of six solitary cutaneous NFs observed unusual high density of blood vessels in the stroma [63] The number of blood vessels in the stroma was found to be increased (on average, 50.7 per 10 high-power fields) in comparison to the average number of blood vessels (23.4 per 10 high-power fields) in five different classic cNFs. In view of this unusual finding, the authors considered these lesions a new histopathological variant of cNF and proposed the name "Angioneurofibroma" (Figure 12). This further emphasizes the angiogenic potential of benign tumors such as cNF. The lesions in this case series were solitary and were not described in patients with NF1. Further studies are required to establish this novel variant.

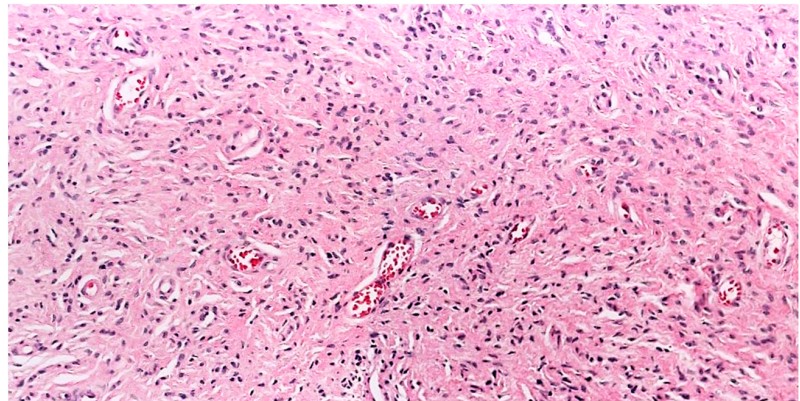

**Figure 12.** Angioneurofibroma, H&E, 100× (courtesy: Dr. Kaya Gürkan).

## 4. Other Potential Histopathological Variants Encountered in our Practice

### 4.1. Plaque

The lesion was seen predominantly in the upper dermis, like a plaque. The cells were arranged in a horizontal plate-like distribution in the dermis (Figure 13). The diagnosis was established with positive staining with S-100 and negative staining with Mart-1.

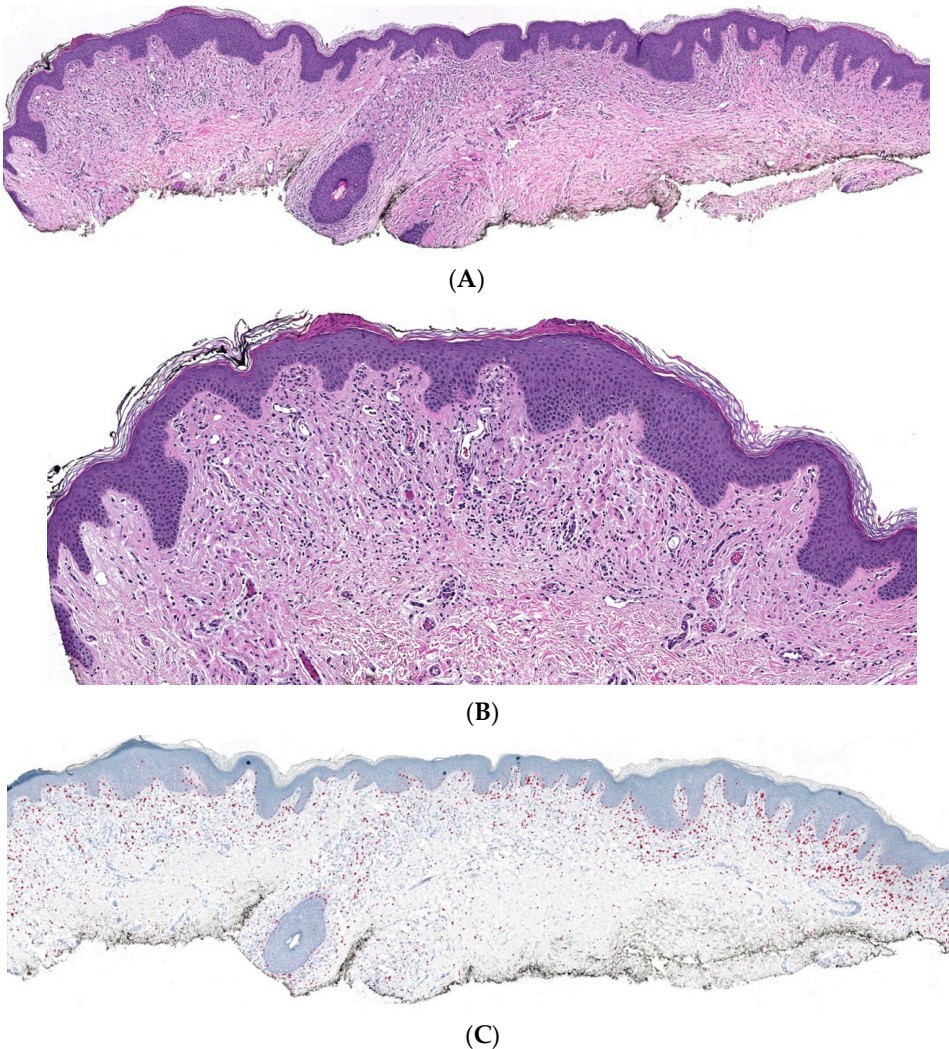

(**A**)

(**B**)

(**C**)

**Figure 13.** Plaque like NF in the upper dermis, H&E, (**A**) 02×, (**B**) 05X, (**C**)-MART-1, 02x. (courtesy: Deon Wolpowitz).

### 4.2. Dispersed

The lesion was ill-defined in the dermis extending into the deep dermis with cells haphazardly arranged (Figure 14). The stroma was loose, not myxoid or collagenous suggestive of conventional NF (Figure 14). The diagnosis was established with positive staining with S-100 and negative staining with Mart-1and p53.

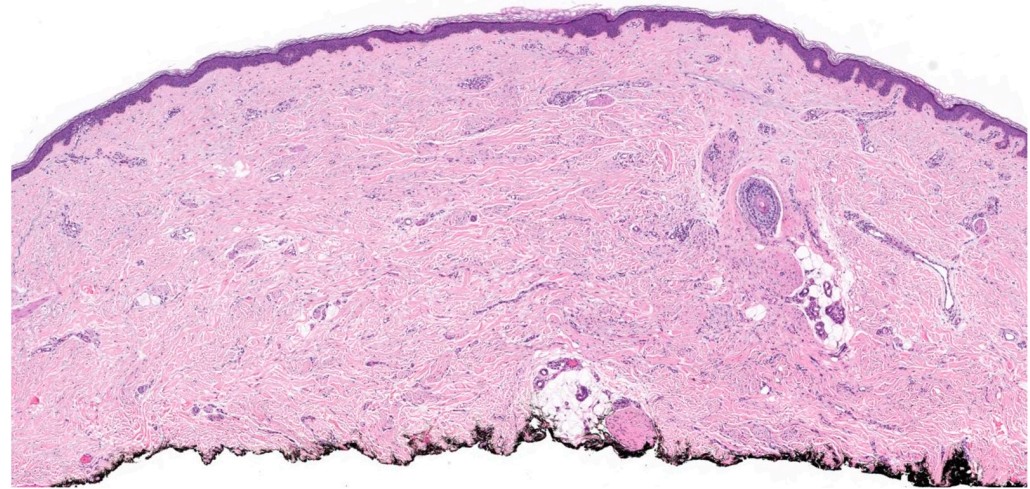

**Figure 14.** Ill-defined lesion composed of haphazardly arranged spindle cells in between collagen bundles, H&E, 20×.

### 5. Conclusions

To summarize, the most common type of cNF is the classic cNF. Variations in the cytomorphology and stromal characteristics in classic cNF give rise to different histopathological subtypes. The identification of areas of classic cNF and immunohistochemistry are helpful in cNF diagnosis, particularly in cases showing atypical features. There is no significant difference in the clinical presentation of different subtypes. Criteria for novel variants are yet to be established. Further exploration is required on the pathogenesis of each variant to understand its biological behavior.

**Author Contributions:** Conceptualization, J.B.; methodology, J.B.; data curation, N.S.N. and J.B.; writing—original draft preparation, N.S.N.; writing—review and editing, N.S.N. and J.B. All authors have read and agreed to the published version of the manuscript.

**Funding:** This research received no external funding.

**Institutional Review Board Statement:** Not applicable.

**Informed Consent Statement:** Not applicable.

**Acknowledgments:** The authors thank Tammie C. Ferringer (Geisinger Medical Center, Danville, PA, USA), Lynne Goldberg (Boston University School of Medicine, Boston, MA, USA), Kaya Gürkan (University Hospital of Geneva, Geneva, Switzerland), Raj Singh (www.pathpresenter.net accessed on 1 October 2022) Mark J. Wilsher (Douglass Hanly Moir Pathology, North Ryde, NSW, Australia), and Deon Wolpowitz (Dermatology and Skin Care Associates, Wellesley, MA, USA), for providing histology pictures of the cases as mentioned in the legends of the figures.

**Conflicts of Interest:** The authors declare no conflict of interest.

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
