# Peer review of "Histopathological Variants of Cutaneous Neurofibroma: A Compendious Review"

_dermatopathology, doi:10.3390/dermatopathology10010001_

Round 1

Reviewer 1 Report

I thank the academic editor for giving me the pleasure of reviewing this interesting paper in which the authors conduct a study on the common and uncommon variants of cutaneous Neurofibroma (cNF). In this way, the authors conduct a very interesting review with the aim to help dermatopathologists to avoid misdiagnosis with other benign and malignant neural tumors.

Introduction: The authors explain well the background in which this manuscript was written and it’s very clear the reasons that make it necessary. It’s fine.

Please, in the caption of figures correct the capital letters (e.g. after the brackets it’s necessary the “capital letter” (line 95 for example).

Regarding Ballon cell/clear cell cNF please discuss shortly differential diagnosis with other (malignant) tumors. I suggest to study, cite, add and discuss this paper:

Cazzato G, Cascardi E, Colagrande A, Cimmino A, Ingravallo G, Lospalluti L, Romita P, Demarco A, Arezzo F, Loizzi V, Dellino M, Trilli I, Bellitti E, Parente P, Lettini T, Foti C, Cormio G, Maiorano E, Resta L. Balloon Cell Melanoma: Presentation of Four Cases with a Comprehensive Review of the Literature. Dermatopathology (Basel). 2022 Mar 28;9(2):100-110. doi: 10.3390/dermatopathology9020013. PMID: 35466242; PMCID: PMC9036264.

Line 265: “Melan-A, HMB-45”: I think that would be better to replace the comma with the word “and”.

Author Response

I thank the academic editor for giving me the pleasure of reviewing this interesting paper in which the authors conduct a study on the common and uncommon variants of cutaneous Neurofibroma (cNF). In this way, the authors conduct a very interesting review with the aim to help dermatopathologists to avoid misdiagnosis with other benign and malignant neural tumors.

Introduction: The authors explain well the background in which this manuscript was written and it’s very clear the reasons that make it necessary. It’s fine.

Please, in the caption of figures correct the capital letters (e.g. after the brackets it’s necessary the “capital letter” (line 95 for example). Done

Regarding Ballon cell/clear cell cNF please discuss shortly differential diagnosis with other (malignant) tumors. I suggest to study, cite, add and discuss this paper:

Cazzato G, Cascardi E, Colagrande A, Cimmino A, Ingravallo G, Lospalluti L, Romita P, Demarco A, Arezzo F, Loizzi V, Dellino M, Trilli I, Bellitti E, Parente P, Lettini T, Foti C, Cormio G, Maiorano E, Resta L. Balloon Cell Melanoma: Presentation of Four Cases with a Comprehensive Review of the Literature. Dermatopathology (Basel). 2022 Mar 28;9(2):100-110. doi: 10.3390/dermatopathology9020013. PMID: 35466242; PMCID: PMC9036264.

The differential diagnosis for clear cell neoplasm are nicely discussed in ref 20. The article suggested by the reviewer is not very relevant to be mentioned in our article. However, we have added a short discusssion and added the refrence

Line 265: “Melan-A, HMB-45”: I think that would be better to replace the comma with the word “and”. Done

Reviewer 2 Report

Nice comprehensive review of neurofibromas. It would be nice to see photomicrographic examples of all the subtypes described rather than just some of them. 

What is the difference between Hyalinized cNF and sclerotic NF?

Author Response

Nice comprehensive review of neurofibromas. It would be nice to see photomicrographic examples of all the subtypes described rather than just some of them.

We totally agree with the reviewer to show more photomicrographs. We requested fron many authors but were unsuccessful. We may get few more and send as addendum.

What is the difference between Hyalinized cNF and sclerotic NF?

The difference between hyalinized Nf and sclerotic NF comes down to basic difference between hyalinization vs sclerosis and is not clear. Ref 52 cites [In keeping with the implication of mast cells as collagen stimulators, our study demonstrates a significantly increased density of mast cells within hyalinized variants of neurofibroma when compared with their classical counterparts. Although, through this observation, causality cannot be definitively concluded, it is compelling evidence that supports the findings of others who have studied the relationship of mast cells with fibrosis and sclerosis at various anatomical sites.] I don’t think the authors try to differentiate. Most of the times the words are used interchangeably. Ref 53 on sclerotic NF [ Histopathologically, the lesion consists of hyalinized thick collagen bundles with prominent clefts characteristically arranged in a plywood-like or whorled pattern].